# Relationship between Tooth Loss and the Medications Used for the Treatment of Rheumatoid Arthritis in Japanese Patients with Rheumatoid Arthritis: A Cross-Sectional Study

**DOI:** 10.3390/jcm10040876

**Published:** 2021-02-20

**Authors:** Hiroko Hashimoto, Shimpei Hashimoto, Yoshihiro Shimazaki

**Affiliations:** 1Department of Preventive Dentistry and Dental Public Health, School of Dentistry, Aichi Gakuin University, Nagoya 464-8650, Japan; hsmt@dpc.agu.ac.jp; 2Hashimoto Orthopedic Surgery Clinic, Nagoya 462-0026, Japan; yt8im3@bma.biglobe.ne.jp

**Keywords:** rheumatoid arthritis, tooth loss, biological therapy, methotrexate, epidemiology

## Abstract

Background: There is limited information regarding the association between tooth loss and the medications used for the treatment of rheumatoid arthritis (RA). Here, we examined the association between tooth loss, disease severity, and drug treatment regimens in RA patients. Method: This study recruited 94 Japanese patients with RA. The severity of RA was assessed using the Steinbrocker classification of class and stage. Data on RA medications were obtained from medical records. We examined the associations between tooth loss, RA severity, and drug treatment regi mens using multinomial logistic regression analyses. Results: Patients with 1–19 teeth had significantly higher odds ratios (ORs) of taking methotrexate (MTX) (OR, 8.74; 95% confidence interval (CI), 1.11–68.8) and biologic disease-modifying antirheumatic drugs (bDMARDs) (OR, 21.0; 95% CI, 1.3–339.1) compared to those with 27–28 teeth when adjusted for RA severity (class). Furthermore, patients with 1–19 teeth had significantly higher ORs of taking MTX (OR, 9.71; 95% CI, 1.22–77.1) and bDMARDs (OR, 50.2; 95% CI, 2.55–990.6) compared to those with 27–28 teeth when adjusted for RA severity (stage). Conclusion: RA patients with fewer teeth were more likely to take stronger RA therapies, independent of RA severity and other factors.

## 1. Introduction

Rheumatoid arthritis (RA) is an inflammatory disease characterized by lesions of the joint synovium. RA affects 0.5–1.0% of the population in Japan. With a male/female ratio of 1/4, new cases are most commonly diagnosed in females aged 30–60. The progression of RA is characterized by destruction of the affected bone and articular cartilage, along with deformity and dislocation of the joints, resulting in chronic dysfunction due to the progression of joint damage. Indeed, it has been suggested that juvenile idiopathic arthritis could affect the temporomandibular joint of patients [1,2,3]. The primary goals of RA treatment are therefore focused on improving the patient’s physical and mental well-being via pain reduction, prevention of joint destruction, and maintenance of function.

Pharmaceutical intervention remains one of the most common treatments for RA [4]. Common RA drugs include nonsteroidal anti-inflammatory drugs (NSAIDs), disease-modifying antirheumatic drugs (DMARDs), methotrexate (MTX), and biologic disease-modifying antirheumatic drugs (bDMARDs). As RA treatment modalities are determined based on clinical factors such as RA severity and disease activity [5], patients with severe RA symptoms are more likely to be treated with strong RA drugs such as bDMARDs.

Recent studies have reported a significant association between periodontitis and RA [6,7], with RA patients with periodontitis exhibiting a greater degree of RA disease activity [8]. Therefore, periodontal status may affect the choice of RA medications. Furthermore, because the progression of periodontitis is strongly associated with tooth loss [9], patients with tooth loss may have suffered from severe periodontitis in the past, which may in turn affect RA treatment strategies. In previous studies examining the relationships between oral factors and RA medications, bDMARDs were reported to improve not only the symptoms of RA but also inflammation of the periodontal tissue [10,11]. On the other hand, few studies have investigated the relationship between tooth loss and RA medications [12]. In this study, we investigate the association between tooth loss and RA treatment strategies by examining the relationship between the number of teeth, RA severity, and RA medication status in RA patients.

## 2. Materials and Methods

### 2.1. Study Population

A total of 94 Japanese adults with RA (mean age, 62.3 ± 14.2 years) were recruited from an orthopedic clinic in Aichi Prefecture, Japan from April 2015 to March 2016. This investigation was performed as a cross-sectional observational study. All patients fulfilled the 1987 revised classification criteria of the American Rheumatism Association [13]. For inclusion and exclusion criteria, inclusion in this study was limited to patients aged ≥18 years, with both edentulous and pregnant patients being excluded. This study was approved by the Ethics Committee of Aichi Gakuin University, School of Dentistry (approval number 405) and conducted in full accordance with the Declaration of Helsinki. Informed written consent was obtained from all patients prior to inclusion in this study.

### 2.2. Assessment of Clinical Rheumatological Parameters

RA patients were characterized using the Steinbrocker functional class and radiographic stage [14]. RA drug treatment regimens were determined based on patient medical records and divided into five categories: NSAIDs, corticosteroids, DMARDs, MTX, and bDMARDs.

### 2.3. Oral Examination

The oral health status of all participants was examined by a single dentist at the orthopedic clinic blinded to the clinical status of the subjects. The number of remaining teeth for each patient was counted, excluding third molars. As a parameter of periodontal health status, probing pocket depth (PPD) was assessed at six points (mesiobuccal, mid-buccal, distobuccal, mesiolingual, mid-lingual, and distolingual) around all teeth using a periodontal probe (CPUNC 15, Hu-Friedy, Chicago, IL, USA). The PPD was recorded to the nearest millimeter, and every observation close to 0.5 mm was rounded to the lowest whole number. Periodontal examiner reliability was verified using an intra-examiner calibration of four volunteers; the percentage of agreement (within ±1 mm) ranged from 81.0% to 98.8% for the PPD. The kappa value ranged from 0.69 to 0.98.

### 2.4. Questionnaires

Data regarding height, body weight, and lifestyle factors were collected using a self-administered questionnaire. Body mass index (BMI; kg/m^2^) was calculated using the patient’s height and body weight. Smoking status was classified as never, former, or current smoker. A former smoker was defined as a patient who had smoked before, but who did not smoke at the time of the investigation.

### 2.5. Statistical Analyses

The RA evaluation indices were divided into three subcategories for class and stage (I, II, and III–IV). RA drug treatment was classified into three categories: bDMARDs, MTX, and other. The number of teeth was divided into three categories (1–19, 20–26, and ≥27 teeth). We used the mean value of all PPD measurements in the analysis. The Kruskal–Wallis test and chi-square test were used to analyze the association between the number of teeth, RA medications, and other variables. Multinomial logistic regression analysis was performed to calculate the odds ratios (ORs) and the 95% confidence intervals (CIs) for the effects of tooth loss and other variables on RA medication. Age, sex, BMI, number of teeth, mean PPD, smoking status, and RA severity (class and stage) were included as independent variables in the multivariate multinomial logistic regression analyses. In the multivariate analyses, we used two models. Model 1 included stage and model 2 included class as independent variables for assessing RA severity. A *p*-value < 0.05 was regarded as indicative of statistical significance. SPSS ver. 23.0 (IBM Japan, Tokyo, Japan) was used for the analyses.

## 3. Results

Patient characteristics are shown in Table 1. Among the noteworthy characteristics seen in this cohort, 24.5% of patients were treated using bDMARDs, and 29.8% of patients had 1–19 teeth.

The association between tooth loss and other variables in RA patients is shown in Table 2. Older patients exhibited a tendency to have fewer teeth, as did those with higher mean PPD, more severe RA, and more extensive RA medication.

The association between RA medication and other variables in RA patients is shown in Table 3. Those with fewer teeth, females, and those with more severe RA (class and stage) displayed a tendency towards more extensive RA medication.

The results of multivariate multinomial logistic regression analyses comparing tooth loss and other variables with RA medication are shown in Table 4. Patients with 1–19 teeth were exhibited a significantly higher frequency of MTX (OR, 8.74; 95% CI, 1.11–68.8) and bDMARDs (OR, 21.0; 95% CI, 1.3–339.1) treatment compared to those with 27–28 teeth when adjusted for RA severity (class). Furthermore, patients with 1–19 teeth had significantly higher ORs of MTX (OR, 9.71; 95% CI, 1.22–77.1) and bDMARD (OR, 50.2; 95% CI, 2.55–990.6) treatment compared to those with 27–28 teeth when adjusted for RA severity (stage).

## 4. Discussion

The data presented here suggest that RA patients with fewer teeth were more likely to have taken bDMARDs, independent of RA severity and other factors. This should be taken into consideration not only for the loss of multiple teeth but also for the possibility that as an important factor for tooth loss, periodontal disease precedes the loss of teeth.

RA patients with moderate and severe disease activity were previously shown to exhibit more extensive tooth loss compared to those in remission or those with low disease activity [15]. Similarly, RA patients with high disease activity scores based on C-reactive protein (DAS28–CRP) levels were more likely to exhibit significant tooth loss [16]. In this study, patients with severe RA tended to have fewer teeth and were more likely to have been treated with bDMARDs, independent of RA severity. According to the RA treatment guidelines, MTX is recommended when disease activity does not improve in response to anti-rheumatic drugs, after which bDMARDs are recommended when disease activity does not improve in response to MTX [5]. Our observation that RA patients with fewer teeth were more likely to have been treated with bDMARDs independent of RA severity suggests that tooth loss is a negative predictor of RA symptom relief. Alongside serum data, radiographs, and other tests necessary to diagnose RA, a review of the patient’s oral status, including the number of teeth, may be helpful in estimating the prognosis of RA treatment outcomes.

The relationship between periodontitis and RA is well established [6,7]. As RA patients with periodontitis are more likely to exhibit more severe RA symptoms than to those without periodontitis, periodontitis may be an independent factor influencing the severity of RA [8]. As worsening periodontitis increases the risk of tooth loss [9], the loss of multiple teeth as a result of disease progression may have an impact on the severity of RA. In the present study, patients with fewer teeth also had deeper periodontal pockets, suggesting that they were losing teeth due to periodontal disease. Hence, periodontitis may have influenced the severity of RA in these patients. On the other hand, the severity of RA may worsen the effects of periodontitis. Although periodontal status was not significantly associated with bDMARDs, the presence of advanced periodontal disease may have contributed to the more severe RA symptoms and associated treatments used in these patients. Although not all tooth loss in this study population can be definitively linked to periodontal disease, periodontal disease remains the most common cause of tooth loss in adulthood [17]. These results suggest that controlling the severity of periodontitis may also reduce the severity of RA, enabling greater treatment response and symptom relief with milder RA drugs. The encouragement of more frequent dental checkups in RA patients may prove beneficial for controlling the progression of periodontitis, which in turn may reduce the severity of RA.

Bone destruction in the oral cavity due to RA is commonly seen in the temporomandibular joint and alveolar bone. When RA symptoms spill over into the temporomandibular joint, it is typically accompanied by inflammation, resulting in resorption, destruction, and deformation on both sides of the temporomandibular joint heads [18,19]. Furthermore, patients with high levels of anti-citrullinated protein antibody (ACPA), an antibody specific to RA patients, exhibited more advanced alveolar bone resorption [20]. RA drugs can provide symptom relief by inhibiting RA-associated arthritis and suppressing the progression of bone destruction [21,22,23]. However, in a previous study examining the effects of RA drugs on periodontal tissue, RA drugs reduced gingival swelling and redness but did not inhibit alveolar bone resorption [11]. While RA drugs may be effective in inhibiting arthritis and bone destruction, they may not be effective for inhibiting alveolar bone destruction. Although many patients with significant tooth loss were taking bDMARDs, the results of this study suggest that the medication did not inhibit the underlying alveolar bone resorption that leads to tooth loss.

Decreased chewing ability due to tooth loss can affect food selection and nutrient intake, which in turn increases the risk of malnutrition [24]. Poor nutrition is a well-established risk factor for infectious diseases, including viral infections that have been implicated in the development of RA [25,26]. Therefore, worsening nutritional status due to tooth loss may serve as an indirect risk factor for RA. Furthermore, poor nutrition is also associated with decreased immune function [27]. RA is an autoimmune disease, in which the immune system attacks otherwise healthy host cells and tissues [28]. Tooth loss may lead to an exacerbation of RA symptoms via the dysregulation of immune function, leading to an increase in the number of patients requiring strong anti-RA drugs, such as bDMARDs.

This study had several limitations. It was impossible to determine a causal association between tooth loss, RA severity, and RA medication status due to the lack of long-term follow-up used in this study. As this study targeted a limited number of RA patients drawn from one orthopedic clinic, the distribution of RA medication may have differed from that of the general population. Furthermore, participation in this study was limited to patients with relatively mild RA severity capable of receiving treatment in an outpatient setting and did not include those with severe RA who required hospitalization. Reduced saliva production is a frequent extra-articular manifestation of RA [29] that affects oral health status, including periodontal condition [30]. However, we did not examine the saliva secretion of patients. To clarify the relationship between tooth loss and RA medication, it will be necessary to conduct a more extensive survey of RA patients taking varying RA medications. Further studies should be conducted to ascertain the degree of tooth loss prior to the start of treatment in RA patients, followed by a longitudinal evaluation to clearly define the relationship between tooth loss and RA medication.

## 5. Conclusions

In this study, RA patients taking bDMARDs were more likely to exhibit significant tooth loss compared to those taking other medications, independent of RA severity and other factors. It is unlikely that physicians treating RA consider the degree of tooth loss when prescribing RA medications, suggesting that tooth loss is an independent factor affecting treatment response in RA patients. As this study represents one of the first reports examining the relationship between tooth loss and RA medications, future studies will be needed to clarify the relationship.

## Figures and Tables

**Table 1 jcm-10-00876-t001:** Characteristics of study patients.

Characteristics	Median (25th Percentile, 75th Percentile) or *n* (%)
Age	62.0 (53.8, 73.3)
BMI (kg/m^2^)	21.3 (19.3, 23.2)
Mean PPD (mm)	2.85 (2.52, 3.34)
Sex	
Male	27 (28.7)
Female	67 (71.3)
Smoking status	
Never	53 (56.4)
Former	20 (21.3)
Current	21 (22.3)
Number of teeth	
27–28	34 (36.2)
20–26	32 (34.0)
≤19	28 (29.8)
RA severity (class)	
Ⅰ	65 (69.1)
Ⅱ	17 (18.1)
III–IV	12 (12.8)
RA severity (stage)	
Ⅰ	56 (59.6)
Ⅱ	19 (20.2)
III–IV	19 (20.2)
RA medication	
NSAIDs/Corticosteroids/DMARDs	32 (34.0)
MTX	39 (41.5)
bDMARDs	23 (24.5)

BMI, body mass index; PPD, probing pocket depth; RA, rheumatoid arthritis; NSAIDs, nonsteroidal anti-inflammatory drugs; DMARDs, disease-modifying antirheumatic drugs; MTX, methotrexate; bDMARDs, biologic disease-modifying antirheumatic drugs.

**Table 2 jcm-10-00876-t002:** Relationship between number of teeth and other variables in patients with RA.

	Number of Teeth			
	27–28	20–26	≤19	
	*n* = 34	*n* = 32	*n* = 28	*p*-Value
Age	51.5 (44.0, 61.3)	62.5 (57.0, 71.0)	73.0 (65.3, 81.8)	<0.001
BMI (kg/m^2^)	20.3 (18.8, 22.3)	22.4 (19.7, 24.5)	21.3 (19.4, 23.3)	0.07
Mean PPD (mm)	2.57 (2.48, 2.77)	2.79 (2.49, 3.27)	3.41 (2.92, 4.24)	<0.001
Sex				
Male	9 (26.5)	10 (31.3)	8 (28.6)	0.91
Female	25 (73.5)	22 (68.8)	20 (71.4)	
Smoking status				
Never	21 (61.8)	16 (50.0)	16 (57.1)	0.05
Former	2 (5.9)	11 (34.4)	7 (25.0)	
Current	11 (32.4)	5 (15.6)	5 (17.9)	
RA severity (class)				
Ⅰ	29 (85.3)	21 (65.6)	15 (53.6)	0.009
II	3 (8.8)	9 (28.1)	5 (17.9)	
III or IV	2 (5.9)	2 (6.3)	8 (28.6)	
RA severity (stage)				
Ⅰ	20 (58.8)	20 (62.5)	16 (57.1)	0.94
II	8 (23.5)	5 (15.6)	6 (21.4)	
III or IV	6 (17.6)	7 (21.9)	6 (21.4)	
RA medication				
NSAIDs/ Corticosteroids/ DMARDs	17 (50.0)	11 (34.4)	4 (14.3)	0.01
MTX	11 (32.4)	16 (50.0)	12 (42.9)	
bDMARDs	6 (17.6)	5 (15.6)	12 (42.9)	

Data are presented as median (25th percentile, 75th percentile) or number of subjects (%). RA, rheumatoid arthritis; BMI, body mass index; PPD, probing pocket depth; NSAIDs, nonsteroidal anti-inflammatory drugs; DMARDs, disease-modifying antirheumatic drugs; MTX, methotrexate; bDMARDs, biologic disease-modifying antirheumatic drugs.

**Table 3 jcm-10-00876-t003:** Relationship between A medication and other variables in patients with RA.

	RA Medication			
	NSAIDs/ Corticosteroids/ DMARDs	MTX	bDMARDs	
	*n* = 32	*n* = 39	*n* = 23	*p*-Value
Age	60.5 (52.3, 66.7)	62.0 (54.0, 73.0)	68.0 (57.0, 83.0)	0.07
BMI (kg/m^2^)	20.8 (18.8, 23.7)	21.6 (19.4, 23.5)	21.4 (19.6, 22.9)	0.74
Mean PPD (mm)	2.68 (2.47, 3.15)	2.94 (2.64, 3.64)	2.85 (2.52, 3.44)	0.17
Number of teeth				
27–28	17 (53.1)	11 (28.2)	6 (26.1)	0.01
20–26	11 (34.4)	16 (41.0)	5 (21.7)	
≤19	4 (12.5)	12 (30.8)	12 (52.2)	
Sex				
Male	12 (37.5)	13 (33.3)	2 (8.7)	0.05
Female	20 (29.9)	26 (66.7)	21 (91.3)	
Smoking status				
Never	16 (50.0)	20 (51.3)	17 (73.9)	0.35
Former	7 (21.9)	9 (23.1)	4 (17.4)	
Current	9 (28.1)	10 (25.6)	2 (8.7)	
RA severity (class)				
I	28 (87.5)	29 (74.4)	8 (34.8)	<0.001
II	3 (9.4)	7 (17.9)	7 (30.4)	
III or IV	1 (3.1)	3 (7.7)	8 (34.8)	
RA severity (stage)				
I	26 (81.3)	25 (64.1)	5 (21.7)	<0.001
II	4 (12.5)	9 (23.1)	6 (26.1)	
III or IV	2 (6.3)	5 (12.8)	12 (52.2)	

Data are presented as median (25th percentile, 75th percentile) or number of subjects (%). RA, rheumatoid arthritis; NSAIDs, nonsteroidal anti-inflammatory drugs; DMARDs, disease-modifying antirheumatic drugs; MTX, methotrexate; bDMARDs, biologic disease-modifying antirheumatic drugs; BMI, body mass index; PPD, probing pocket depth.

**Table 4 jcm-10-00876-t004:** Association of the number of teeth and other variables with RA medication by multivariate multinomial logistic regression analyses.

	Model 1		Model 2	
	Dependent Variable: RA Medication		Dependent Variable: RA Medication	
	MTX vs. NSAIDs/Corticosteroids/ DMARDs	bDMARDs vs. NSAIDs/Corticosteroids/ DMARDs	MTX vs. NSAIDs/Corticosteroids/ DMARDs	bDMARDs vs. NSAIDs/Corticosteroids/ DMARDs
Independent variable	Adjusted OR (95% CI)	Adjusted OR (95% CI)	Adjusted OR (95% CI)	Adjusted OR (95% CI)
Age	0.96 (0.91, 1.01)	0.95 (0.89, 1.02)	0.96 (0.92, 1.01)	0.99 (0.93, 1.05)
BMI (kg/m^2^)	1.10 (0.93, 1.30)	1.19 (0.96, 1.47)	1.06 (0.89, 1.26)	1.10 (0.88, 1.37)
Mean PPD (mm)	1.05 (0.34, 3.28)	0.47 (0.10, 2.28)	1.22 (0.39, 3.75)	0.35 (0.07, 1.78)
Number of teeth				
27–28	1	1	1	1
20–26	2.95 (0.72, 12.1)	0.97 (0.14, 6.85)	3.66 (0.88, 15.1)	1.55 (0.20, 12.2)
≤19	8.74 (1.11, 68.8) *	21.0 (1.30, 339.1) *	9.71 (1.22, 77.1) *	50.2 (2.55, 990.6) *
Sex				
Male	1	1	1	1
Female	1.31 (0.37, 4.68)	5.56 (0.69, 45.2)	1.28 (0.36, 4.54)	5.41 (0.68, 43.2)
Smoking status				
Never	1	1	1	1
Former	0.82 (0.18, 3.68)	0.97 (0.13, 7.44)	0.69 (0.15, 3.14)	1.14 (0.14, 9.03)
Current	0.84 (0.21, 3.40)	0.89 (0.10, 7.59)	0.69 (0.16, 2.86)	0.96 (0.09, 9.84)
RA severity (class)				
I	1	1		
II	2.54 (0.51, 12.7)	15.6 (2.19, 111.1) **		
III or IV	4.47 (0.27, 73.7)	62.4 (2.95, 1320.9) **		
RA severity (stage)				
I			1	1
II			3.44 (0.84, 14.1)	8.32 (1.20, 57.5) *
III or IV			2.23 (0.33, 15.2)	59.7 (6.86, 519.1) **

RA, rheumatoid arthritis; MTX, methotrexate; NSAIDs, nonsteroidal anti-inflammatory drugs; DMARDs, disease-modifying antirheumatic drugs; bDMARDs, biologic disease-modifying antirheumatic drugs; OR, odds ratio; CI, confidence interval; BMI, body mass index; PPD, probing pocket depth. * *p* < 0.05; ** *p* < 0.01.

## Data Availability

The datasets generated during and/or analyzed during the current study are available from the corresponding author on reasonable request.

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
