# Peer review of "Relationship between Tooth Loss and the Medications Used for the Treatment of Rheumatoid Arthritis in Japanese Patients with Rheumatoid Arthritis: A Cross-Sectional Study"

_jcm, 2021, doi:10.3390/jcm10040876_

Round 1
Reviewer 1 Report
Dear Authors, thank you for submitting your paper.
The aim of the present study was to evaluate the association between tooth loss, disease severity, and drug treatment regimens in RA patients.
I congratulate the authors for this very relevant research, which will add to the dental field.
The study is interesting and It appears well structured, correctly carried out and written without logical or factual errors
Methodological aspects are deeply cleared in the manuscript.
The topic is in line with the journal aim.
As rheumatoid arthritis could affect the temporomandibular joint(TMJ) of the patients I suggest to the authors to improve their introduction citing and discussing the following research:
https://doi.org/10.3390/children8010033
https://doi.org/10.1007/s00784-019-03122-5
https://doi.org/10.3390/jcm9041159
-Data reported in the Methods section are appropriate and precisely described;.
-Results are reported clearly and adequately supported by Tables.
The Conclusions are correctly stated and supported by the findings obtained from the present study. The authors stated that RA patients taking bDMARDs were more likely to exhibit significant tooth loss compared to those taking other medications, independent of RA severity and other factors.
The present findings could be very useful for physicians, as stated by the authors it is important for them to consider the degree of tooth loss when they prescribe RA medications
According to this Reviewer’s consideration, novelty and quality of the paper, publication of the present manuscript is recommended.
Reviewer 2 Report
The study is A Cross-Sectional which links tooth loss and RA as well as drug therapies associated with them
The study was conducted with correct methodology, and the results are clearly described.
Only a few considerations need to be made:
"The data presented here suggest that RA patients with fewer teeth were more likely to have taken bDMARDs, independent of RA severity and other factors" could depend on the fact that the tooth loss associated with periodontal disease is advanced progressing as the symptomatology is reduced following all hiring the famaci bDMARDs?
In the discussion you affirm the following sentence "Hence, it is possible that periodontitis may have influenced the severity of RA in these patients" but could it also be the opposite that severity RA influenced periodontitis with a worsening of periodontal data?
I consider the manuscript worthy of publication. the incidence of RA and periodontal disease ,as the main cause of loss of dental elements ,are very high in population and any clinical study conducted with correct methodology that relates these 2 pathologies must be published
